# COVID-19 mortality in the United States: It's been two Americas from the start

**Michael A. Stoto⊙\*, Samantha Schlageter⊙, John D. Kraemer**

Department of Health Systems Administration, Georgetown University, Washington, D.C., United States of America

⊙ These authors contributed equally to this work.
\* stotom@georgetown.edu

**Data Availability Statement:** The data underlying the results presented in the study are available from the U.S. Centers for Disease Control and Prevention at https://www.cdc.gov/nchs/nvss/vsrr/covid19/excess_deaths.htm.

## Abstract

During the summer of 2021, a narrative of "two Americas" emerged: one with high demand for the COVID-19 vaccine and the second with widespread vaccine hesitancy and opposition to masks and vaccines. We analyzed "excess mortality" rates (the difference between total deaths and what would have been expected based on earlier time periods) prepared by the CDC for the United States from January 3, 2020 to September 26, 2021. Between Jan. 3, 2020 and Sept. 26, 2021, there were 895,693 excess deaths associated with COVID-19, 26% more than reported as such. The proportion of deaths estimated by the excess mortality method that was reported as COVID-19 was highest in the Northeast (92%) and lowest in the West (72%) and South (76%). Of the estimated deaths, 43% occurred between Oct. 4, 2020 and Feb. 27, 2021. Before May 31, 2020, approximately 56% of deaths were in the Northeast, where 17% of the population resides. Subsequently, 48% of deaths were in the South, which makes up 38% of the population. Since May 31, 2020, the South experienced COVID-19 mortality 26% higher than the national rate, whereas the Northeast's rate was 42% lower. If each region had the same mortality rate as the Northeast, more than 316,234 COVID-19 deaths between May 31, 2020 and Sept. 26, 2021 were "avoidable." More than half (63%) of the avoidable deaths occurred between May 31, 2020 and February, 2021, and more than half (60%) were in the South. Regional differences in COVID-19 mortality have been strong throughout the pandemic. The South has had higher mortality rates than the rest of the U.S. since May 31, 2020, and experienced 62% of the avoidable deaths. A comprehensive COVID-19 policy, including population-based restrictions as well as vaccines, is needed to control the pandemic.

## Introduction

During the summer of 2021, as vaccine uptake slowed, a narrative of "two Americas" emerged: one with a high demand for the COVID-19 vaccine and the second with high vaccine hesitancy, and later widespread opposition to mask and vaccine mandates. The second America was mostly concentrated in Southern states and rural areas, especially in counties that voted for Donald Trump. Through the summer, the number of cases, hospitalizations, and deaths

**Funding:** The author(s) received no specific funding for this work.

**Competing interests:** The authors have declared that no competing interests exist.

increased dramatically in the second America [1, 2]. This narrative shapes not only our understanding of what happened, but also what should, or could, be done to control the ongoing pandemic and future outbreaks.

However, this narrative is not quite true. In fact, our analysis of how COVID-19 mortality evolved over time shows that stark regional differences existed from the start of the pandemic, both in cases and deaths as well as testing and vaccine uptake. To see this, we analyzed "excess mortality," the difference between total deaths and what would have been expected based on earlier time periods. Reported cases, hospitalizations, and deaths are known to substantially underestimate actual infections and deaths by a variable fraction depending on testing availability, patient and physician awareness and attitudes, hospital resources, and other factors [3, 4]. Since COVID-19 awareness and concern [2, 5, 6], as well as test availability and use [7–9], vary markedly throughout the U.S., excess mortality estimates can help avoid potential bias due to systematic differences in reporting between areas with high and low prevalence areas.

## Methods

In order to identify when regional differences in the pandemic emerged, and their temporal association with vaccine uptake, we analyzed the evolution of COVID-19 mortality patterns. Our analysis is based on state-level weekly excess mortality calculations published by CDC [10]. Farrington surveillance algorithms, which use over-dispersed Poisson generalized linear models with spline terms to model trends in counts, accounting for seasonality, were implemented for each jurisdiction (states, plus the District of Columbia and New York City). These models generate a set of expected counts of deaths by week and jurisdiction, and excess mortality is simply the difference between observed and predicted deaths in each week and jurisdiction. Additional details, including weighting to adjust for potential underreporting in the most recent weeks, are provided by CDC [10].

Estimates from other sources might be slightly different, but the differences are not so great as to affect the overall conclusions. In order to avoid reporting lags that potentially vary by region (especially during the summer of 2021 when some states experienced far greater caseloads than others), we concluded the analysis on September 26, 2021. In addition, baseline forecasts, and hence estimated excess mortality, become less reliable further from the pre-pandemic period.

We grouped the states into the four standard Census regions. Other *ad hoc* groupings could exaggerate or minimize differences. The on-line appendix includes figures comparing the per-capita weekly rates of excess mortality to demonstrate the similarity of patterns within each region. This is especially true in the Northeast and South, but there is more internal variation in the West, where California is somewhat of an outlier. Based on a preliminary analysis of the data, we designated five periods with substantially different patterns of excess mortality. To account for differences in population sizes and length of the time periods, calculations were done by week and state, and aggregated into regional and period.

To calculate "avoidable mortality," we subtracted from the observed excess deaths the excess deaths that would have been experienced in each region and period if every region had the lowest excess mortality rate experienced by any region in that period. We did not include the first period in these calculations for the reason described below, but performed a sensitivity analysis in which we estimated the numbers of deaths that could have been avoided in Period 1 as one-half of the difference between the actual rates and those in the region with the lowest excess mortality rate in that period.

Vaccine coverage rates were calculated by Eva Rest of Georgetown University based on a data set compiled by Tiu and colleagues [11].

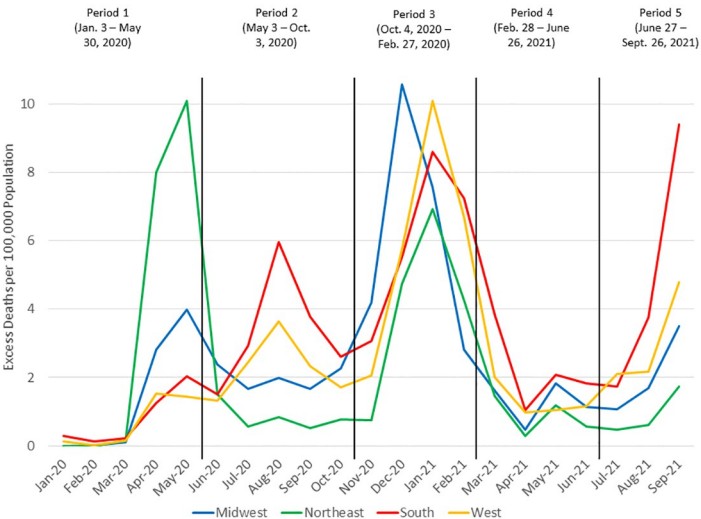

**Fig 1. Excess mortality per 100,000 population by week and region, U.S., January 3, 2020 –September 26, 2021.**
The periods are denoted by vertical lines corresponding to May 30, 2020, Oct. 3, 2020, Feb. 27, 2021, and June 25, 2021.
Source: authors' calculations based on CDC data.

## Results

Between Jan. 3, 2020 and Sept. 26, 2021, the excess mortality associated with COVID-19 in the U.S. totaled 895,693, a per capita rate of 270 per 100,000 population. During the same period, 710,999 COVID-19 deaths were reported, amounting to 79% of the estimated excess mortality. In other words, during this period the U.S. experienced 26% more COVID-19 deaths than reported as such.

As can be seen in Fig 1, over time there are clear differences among the regions, especially between the Northeast and South. Fig 2 shows 43% of the excess deaths occurred between Oct. 4, 2020 and Feb. 27, 2021. Before May 31, 2020 (period 1), approximately 56% of deaths were in the Northeast; subsequently, 48% were in the South.

Controlling for the different sizes of the regions and lengths of the periods, Table 1 displays excess mortality by region per 100,000 population per day. Before May 31, 2020, the daily mortality rate in the Northeast (0.881 per 100,000) was 3.3 times the national rate while the rate for the South (0.132 per 100,000) was 48% of the national rate. Subsequently, the South experienced COVID-19 mortality 26% higher than the national rate, whereas the Northeast's rate was 42% lower.

Table 1 also shows that the ratio of estimated to reported COVID-19 deaths varies over time and among the regions. The proportion is lowest in the West (71.8%) and South (76.2%). The proportions were similar in all regions (approximately 82%) in Period 1, but vary markedly afterward, dropping to less than 60% in the West in some periods. The overall proportion is highest in the Northeast (92%), and is greater than 100% in the third and fourth period (109% and 121% respectively. This is probably because some people who died of another primary cause had a positive COVID-19 test and were included in the reported counts [12], as called for by the National Center for Health Statistics [13]. A recent analysis attributed the more accurate coding of COVID-19 deaths in the New England states (which are in the Northeast region) to well-run and funded public health departments, excellent hospitals, and state medical examiners who ensure death certificate information is both accurate and timely

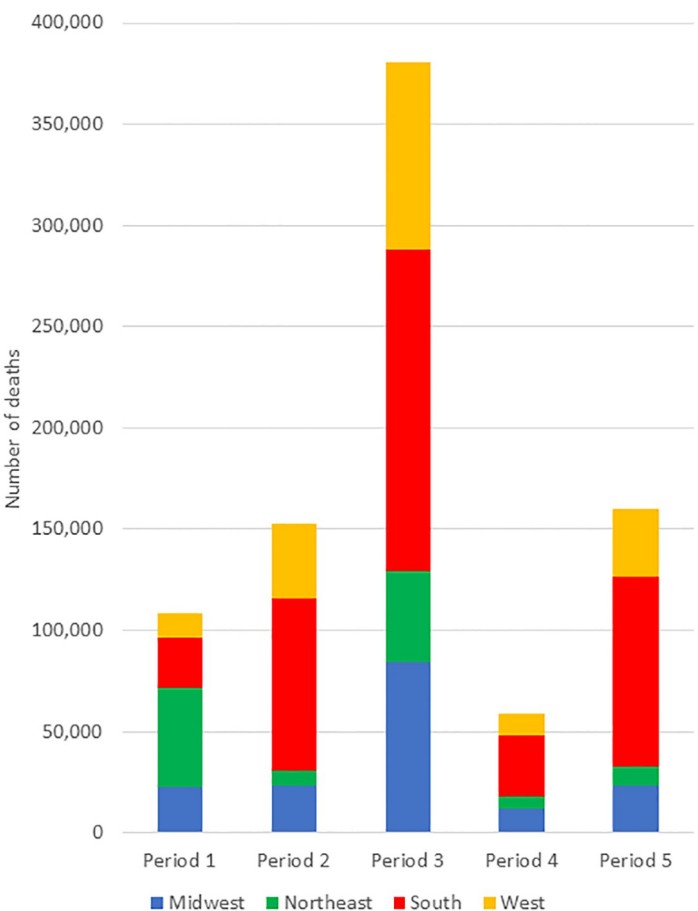

**Fig 2. Number of excess deaths by period and region, U.S., January 3, 2020—September 26, 2021.** The height of each bar represents the total U.S. excess deaths in each period and the colored sections represent the number of excess deaths in each region during the period. Periods are defined as follows: Period 1 (January 3 to May 30, 2020), Period 2 (May 31 –October 3, 2020), Period 3 (October 4, 2020 to February 27, 2021), Period 4 (February 28 to June 26, 2021), and Period 5 (June 27 to September 26, 2021). Source: authors' calculations based on CDC data.

[14]. The report also noted that some states in the region had large increases in deaths from overdoses rather than COVID-19, indicating a weakness of the excess mortality approach.

Because the rates of excess mortality vary so markedly, one can calculate how many deaths could have been avoided if each region had the same rates as the lowest region in each time period. We assume that no deaths were avoidable in the first period, before much was known about treating or preventing COVID-19. Subsequently, we counted as "avoidable" the difference between the rates seen and those in the Northeast, which were the lowest in each period. According to this calculation, 316,234 COVID-19 deaths between May 31, 2020 and Sept. 26, 2021 were "avoidable." Fig 3 shows that more than half of the avoidable deaths (62%) were in the South. More than half (63%) of the avoidable deaths occurred between May 31, 2020 and February, 2021, and an additional 36% between June 27 and Sept. 26, 2021.

## Discussion

The theme of "two Americas" arose in the summer of 2021, regarding at first vaccine refusal, and later opposition to vaccine and mask mandates and more generally, Covid denialism [15].

Using the proportion voting for Donald Trump as a proxy for party affiliation, journalistic analyses found consistently lower vaccination rates and higher COVID-19 mortality in Southern states, which are predominantly Republican, and the opposite in the Northeastern states, which are predominantly Democratic [2, 5, 6]. But there are other geographic differences, including age distribution and education levels, that can be confounding factors [16].

Our analysis of excess mortality demonstrates that large disparities have existed since the beginning of the pandemic in the U.S. The starkest contrast is between the Northeast (which is heavily Democratic) and South (predominantly Republican). As first noted by Woolf and colleagues [17] for 2020, the first wave of the pandemic was highly concentrated in the Northeast, and particularly in the New York metropolitan area. Since May 31, 2020, however, approximately 48% all excess deaths were in the South, which makes up 38% of the population. The disparity was most apparent in the summer of 2020 (May 31 –Oct. 3), when the daily excess mortality rates were 0.539 per 100,000 in the South, 0.369 per 100,000 nationally, and 0.111 per 100,000 in the Northeast. Similarly, of the 316,234 avoidable COVID-19 deaths between May 31, 2020 and Sept. 26, 2021, the majority (62%) were in the South.

As a sensitivity analysis, we estimated the numbers of deaths that could have been avoided in Period 1 as one-half of the difference between the actual rates and those in the West (which had the lowest excess mortality rate in that period). This increases the number of avoidable deaths to 402,700 an increase of 86,466. With this assumption the proportion of avoidable deaths in the South was still high, 56%.

The regional level of analysis is simple enough for disparities to be clearly apparent, but masks more extreme disparities at the state and local levels. A *Washington Post* analysis, for instance, demonstrates a strong relationship between the proportion who voted for Donald Trump in 2020 and COVID-19 mortality at the county-level [18]. A second analysis shows a strong correlation at the state level between vaccine uptake and the Trump vote [18]. Thus, the disparities in this analysis are likely underestimates of actual differences at finer levels of geography.

Strong resistance to vaccine and mask mandates emerged during the summer of 2021, especially in the South and in states with Republican governors [19, 20]. Consistent with this, excess mortality in the summer of 2021 was correlated with vaccine uptake. As seen in Table 1, the Northeast had the lowest COVID-19 mortality rates and the highest vaccination coverage: 52% of the population was fully vaccinated on June 27, 2021 and 60% on September 26, 2021. The South, at the other extreme, had the highest COVID-19 mortality rates and lowest vaccine coverage (40% and 49% on the same dates).

Nationally, vaccines have already saved many lives [21] and boosters have the potential to save many more [22]. However, our analysis suggests that major differences in COVID-19 mortality emerged in the summer of 2020, well before vaccines became available. Indeed, 63% of the avoidable deaths occurred by the end of February, 2021, when the vaccine rollout was just beginning. These avoidable deaths occurred before the Delta variant became dominant in the United States, so Delta is not part of the explanation. Similarly, because the South has had dramatically higher COVID-19 mortality than the Northeast (since June, 2020) during all seasons of the year, weather is not a likely explanation. Northeasters may have carried some natural immunity into the summer of 2020 [23], but by July, 2020, Anand and colleagues estimate substantially higher seroprevalence rates in the South (37.9%) than in the Northeast (17.5%), so natural immunity cannot explain the large differences starting in the summer of 2020 [24]. Excess COVID-19 mortality in the South and other areas of the country, therefore, is likely to be due at least in part to higher transmission resulting from differences in mask use, school attendance, social distancing, and other behaviors.

**Table 1. Excess and avoidable mortality by region and time period, U.S., January 3, 2020 –September 26, 2021.**

| | Period 1 (1/3–5/30/20) | Period 2 (5/31–10/3/20) | Period 3 (10/4/20–2/27/21) | Period 4 (2/28–6/26/21) | Period 5 (6/27–9/26/21) | Entire period |
|---|---|---|---|---|---|---|
| Excess mortality by region and period | | | | | | |
| Midwest | 22,325 | 23,169 | 84,049 | 12,142 | 23,345 | 165,030 |
| Northeast | 75,143 | 7,986 | 51,200 | 7,613 | 10,572 | 152,514 |
| South | 24,667 | 85,031 | 158,350 | 30,270 | 93,712 | 392,030 |
| West | 12,177 | 36,869 | 92,891 | 10,824 | 33,358 | 186,119 |
| U.S. | 134,312 | 153,055 | 386,490 | 60,849 | 160,987 | 895,693 |
| Excess mortality per capita by region and period (per 100,000 population) per day | | | | | | |
| Midwest | 0.219 | 0.227 | 0.823 | 0.119 | 0.229 | 0.381 |
| Northeast | 0.881 | 0.111 | 0.609 | 0.112 | 0.202 | 0.422 |
| South | 0.132 | 0.539 | 0.859 | 0.203 | 0.816 | 0.494 |
| West | 0.105 | 0.375 | 0.810 | 0.117 | 0.466 | 0.377 |
| U.S. | 0.274 | 0.368 | 0.799 | 0.156 | 0.534 | 0.430 |
| Excess mortality as a proportion of reported COVID-19 deaths | | | | | | |
| Midwest | 85.1% | 61.0% | 95.0% | 84.7% | 61.7% | 83.4% |
| Northeast | 82.7% | 98.8% | 108.7% | 121.3% | 55.6% | 92.3% |
| South | 78.8% | 67.1% | 84.6% | 56.0% | 76.2% | 76.2% |
| West | 74.9% | 57.0% | 84.7% | 54.1% | 56.7% | 71.8% |
| U.S. | 81.7% | 65.4% | 90.1% | 69.5% | 68.7% | 79.4% |
| Avoidable excess mortality by region and period | | | | | | |
| Midwest | 0 | 13,606 | 22,738 | 3,026 | 10,685 | 50,055 |
| Northeast | 0 | 0 | 0 | 0 | 0 | 0 |
| South | 0 | 67,528 | 46,131 | 13,584 | 70,541 | 197,783 |
| West | 0 | 25,975 | 23,046 | 439 | 18,936 | 68,395 |
| U.S. | 0 | 107,108 | 91,915 | 17,048 | 100,162 | 316,234 |
| Proportion fully vaccinated by region at the end of the period (% of the entire population) | | | | | | |
| Midwest | | | 7.3% | 44.4% | 52.9% | |
| Northeast | | | 6.8% | 51.9% | 60.3% | |
| South | | | 7.5% | 42.9% | 49.8% | |
| West | | | 7.3% | 39.8% | 48.9% | |
| U.S. | | | 7.5% | 47.7% | 56.5% | |

Compared to estimates based on epidemiologic models, the excess mortality methods and assumptions are simple and straightforward. While they cannot explain _why_ some regions had different mortality rates, excess mortality estimates can accurately document when and where COVID-19 occurred. Indeed, Woolf argues that state mortality differentials during the pandemic simply continue a decades-long trend in political differences between Republican and Democratic dominated states [25]. However, although causal patterns are complex and difficult to ascertain, differential implementation of and adherence to stay-at-home orders, mask use, and other non-pharmaceutical interventions seem to be at least a partial explanation for the regional differences in COVID-19 mortality. Thus, our analysis demonstrates the potential impact of population-based restrictions as well as vaccines, indicating that a comprehensive COVID-19 policy is needed to control future pandemics.

This analysis demonstrates the benefits of the excess mortality approach [26]. This method provides objective estimates of COVID-19's impact on mortality that do not rely on test availability, clinical decisions, and reporting processes that can lead to under—and over–counting. The ratio of estimated to reported COVID-19 deaths ranged from 121% (Northeast in period

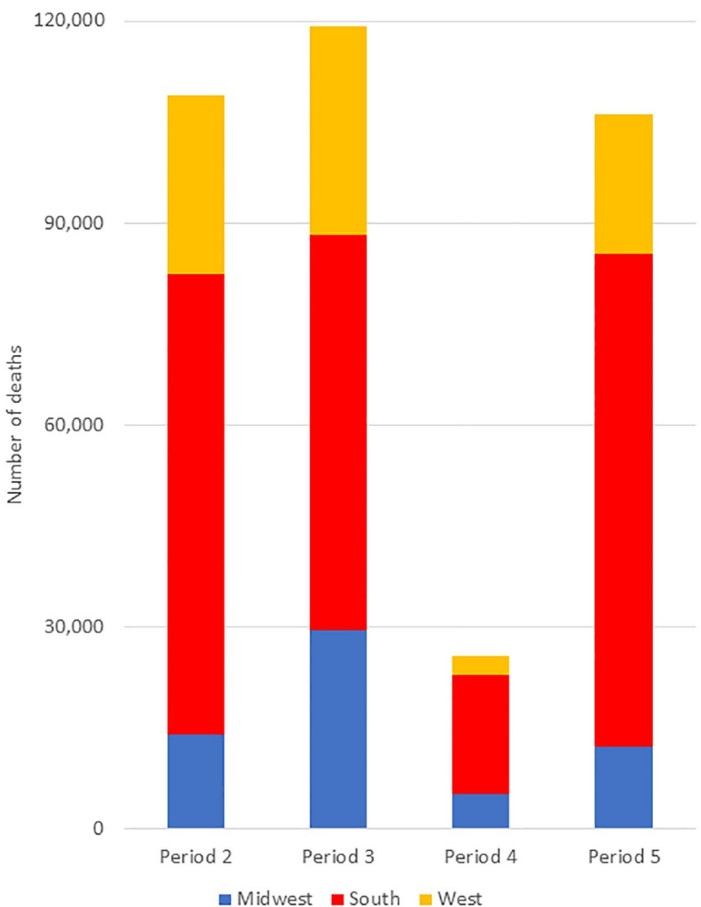

**Fig 3. Number of avoidable deaths by period and region, U.S., January 3, 2020—September 26, 2021.** The height of each bar represents the total U.S. avoidable deaths in each period and the colored sections represent the number of avoidable deaths in each region during the period. Periods defined as follows: Period 1 (January 3 to May 30, 2020), Period 2 (May 31 –October 3, 2020), Period 3 (October 4, 2020 to February 27, 2021), Period 4 (February 28 to June 26, 2021), and Period 5 (June 27 to September 26, 2021). Source: authors' calculations based on CDC data.

4) to 54% in the West and 56% in the South in the same period. An analysis of reported COVID-19 deaths, therefore would have shown both fewer avoidable deaths and regional disparities. The presence of reporting delays, which can be greater in some areas than others, means that excess mortality estimates and differentials are not reliable for a period of weeks, so excess mortality estimates are limited as real-time surveillance tools. And the reliability of these methods will decline as we go further beyond the base period. However, over the period studied, they can be quite informative.

Although the causes are not fully understood, COVID-19 clearly has played out differently across the country over the nearly two years since it emerged. The analysis therefore illustrates how the importance of going beyond cumulative case counts for the U.S. as a whole that dominates the news cycle. Beyond the regional differences that are the focus of this paper, there are substantial differences between rural and urban areas with states and among socio-demographic groups. Because they are less sensitive to differences in reporting patterns than case counts, excess mortality methods can be especially useful in understanding these patterns [27].

## Author Contributions

**Conceptualization:** Michael A. Stoto, Samantha Schlageter.

**Data curation:** Samantha Schlageter.

**Formal analysis:** Michael A. Stoto, Samantha Schlageter, John D. Kraemer.

**Investigation:** Michael A. Stoto, Samantha Schlageter, John D. Kraemer.

**Methodology:** Michael A. Stoto, Samantha Schlageter, John D. Kraemer.

**Supervision:** Michael A. Stoto.

**Validation:** Michael A. Stoto.

**Visualization:** Samantha Schlageter.

**Writing – original draft:** Michael A. Stoto, Samantha Schlageter.

**Writing – review & editing:** John D. Kraemer.

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
