## [Decision Letter · Decision Letter 0]

8 Mar 2022

PONE-D-22-05243COVID-19 mortality in the United States: It's been two Americas from the startPLOS ONE

Dear Dr. Stoto,

Thank you for submitting your manuscript to PLOS ONE. After careful consideration, we feel that it has merit but does not fully meet PLOS ONE’s publication criteria as it currently stands. Therefore, we invite you to submit a revised version of the manuscript that addresses the points raised during the review process.

We look forward to receiving your revised manuscript.

Kind regards,

Wenping Gong, Ph.D.

Academic Editor

PLOS ONE

Journal Requirements:

3. Please remove your figures from within your manuscript file, leaving only the individual TIFF/EPS image files, uploaded separately.  These will be automatically included in the reviewers’ PDF.

Additional Editor Comments:

1. Please present this manuscript as a brief report rather than full article

2. A detaied discription on the section of methods is needed.

3. Please carefully revise the manuscript following the reviewer's comments.

Reviewers' comments:

Reviewer's Responses to Questions

**Comments to the Author**

1. Is the manuscript technically sound, and do the data support the conclusions?

Reviewer #1: Partly

2. Has the statistical analysis been performed appropriately and rigorously? 

Reviewer #1: I Don't Know

3. Have the authors made all data underlying the findings in their manuscript fully available?

Reviewer #1: Yes

4. Is the manuscript presented in an intelligible fashion and written in standard English?

Reviewer #1: Yes

5. Review Comments to the Author

Reviewer #1: The authors are to be congratulated for an insightful analysis highlighting marked differences in excess mortality by state. The manuscript is well-written and, although, is short of novelty it does has value in that it re-iterates the need to consider disaggregated data on excess mortality. There are several aspects of the manuscript in its current format that might benefit from further revisions.

1. There are statements in the Introduction that require referencing, to add substance to inferences (e.g., last sentence).

2. While not a major issue, this Reviewer fails to see the relevance of political context (voting behaviour) for Covid-related excess mortality.

3. Might help unfamiliar readers to have more context as to the rationale for regional differences in excess mortality within the US.

4. In this Reviewer opinion, the Methods section should detail the analytical approach to enable replication of study findings. At a minimum, definition of excess mortality and how differences by region were compared. How did authors adjusted for different region sizes and variation in length of follow-up?

5. Figure are unclear and should include a legend - these should be self-explanatory.

6. The research value of the work could be enhanced by discussing potential reasons for variation in excess mortality across the different regions.

6. PLOS authors have the option to publish the peer review history of their article (what does this mean?). If published, this will include your full peer review and any attached files.

Reviewer #1: No

---

## [Author Response · Author response to Decision Letter 0]

13 Mar 2022

Thank you for the comments we received, and the opportunity to revise and re-submit the paper. Please see our detailed responses below.

Journal Requirements:

We have revised the manuscript to meet the journal’s style requirements

Done

3. Please remove your figures from within your manuscript file, leaving only the individual TIFF/EPS image files, uploaded separately. These will be automatically included in the reviewers’ PDF.

Done

Additional Editor Comments:

1. Please present this manuscript as a brief report rather than full article

The editorial submission system does not give me this option.

2. A detailed description on the section of methods is needed.

The methods section has been expanded as suggested by the Reviewer.

3. Please carefully revise the manuscript following the reviewer's comments.

Please see our responses to the specific comments below.

Review Comments to the Author (Reviewer #1)

The authors are to be congratulated for an insightful analysis highlighting marked differences in excess mortality by state. The manuscript is well-written and, although, is short of novelty it does has value in that it re-iterates the need to consider disaggregated data on excess mortality. There are several aspects of the manuscript in its current format that might benefit from further revisions.

Thank you for these comments. The novelty in this research, we believe, is that it illustrates that regional differences in COVID-19 excess mortality started to mirror deep political differences in the United States long before they became apparent in the uptake of pandemic vaccines. Please see our detailed responses below. 

Please note that we have also made a number of minor edits, mainly updating references, to reflect new publications and a different situation since we originally submitted this manuscript.

1. There are statements in the Introduction that require referencing, to add substance to inferences (e.g., last sentence).

We have added six references to the Introduction, especially the last sentence.

2. While not a major issue, this Reviewer fails to see the relevance of political context (voting behaviour) for Covid-related excess mortality.

As noted above, an important contribution of this research is to illustrate that regional differences in COVID-19 excess mortality started to mirror deep political differences in the United States long before they became apparent in the uptake of pandemic vaccines. In trying to maintain the objectivity that is expected in a scientific journal we may have underplayed these issues, so bearing in mind that non-American readers need more information to interpret this, we have added some text to clarify this point.

3. Might help unfamiliar readers to have more context as to the rationale for regional differences in excess mortality within the US.

Consistent with our response to comment #2, regional differences in the U.S. are driven in large part by political differences, so we have added text on this too.

4. In this Reviewer opinion, the Methods section should detail the analytical approach to enable replication of study findings. At a minimum, definition of excess mortality and how differences by region were compared. How did authors adjusted for different region sizes and variation in length of follow-up?

We have added text to the Methods section on the definition of excess mortality and how it is calculated, how we took into account different region sizes and time periods, and calculated “avoidable mortality.”

5. Figure are unclear and should include a legend - these should be self-explanatory.

We have expanded the legends of the figures to make them more self-explanatory.

6. The research value of the work could be enhanced by discussing potential reasons for variation in excess mortality across the different regions.

As noted above, we believe that the regional differences are driven by political differences, and have added text to make this point.

---

## [Decision Letter · Decision Letter 1]

8 Apr 2022

COVID-19 mortality in the United States: It's been two Americas from the start

PONE-D-22-05243R1

Dear Dr. Michael A Stoto,

We’re pleased to inform you that your manuscript has been judged scientifically suitable for publication and will be formally accepted for publication once it meets all outstanding technical requirements.

Kind regards,

Wenping Gong, Ph.D.

Academic Editor

PLOS ONE

Additional Editor Comments (optional):

Reviewers' comments:

Reviewer's Responses to Questions

**Comments to the Author**

1. If the authors have adequately addressed your comments raised in a previous round of review and you feel that this manuscript is now acceptable for publication, you may indicate that here to bypass the “Comments to the Author” section, enter your conflict of interest statement in the “Confidential to Editor” section, and submit your "Accept" recommendation.

Reviewer #1: All comments have been addressed

Reviewer #2: All comments have been addressed

2. Is the manuscript technically sound, and do the data support the conclusions?

Reviewer #1: Yes

Reviewer #2: Yes

3. Has the statistical analysis been performed appropriately and rigorously? 

Reviewer #1: Yes

Reviewer #2: Yes

4. Have the authors made all data underlying the findings in their manuscript fully available?

Reviewer #1: Yes

Reviewer #2: Yes

5. Is the manuscript presented in an intelligible fashion and written in standard English?

Reviewer #1: Yes

Reviewer #2: Yes

6. Review Comments to the Author

Reviewer #1: No further comments. The authors have satisfactorily addressed my initial comments. Congratulations on an interesting analysis.

Reviewer #2: The authors analyzed excess mortality of COVID-19 in the United States from January 3, 2020 to September 26, 2021. The study is performed well, and the results are straightforward.

7. PLOS authors have the option to publish the peer review history of their article (what does this mean?). If published, this will include your full peer review and any attached files.

Reviewer #1: No

Reviewer #2: No

---

## [Editor Report · Acceptance letter]

19 Apr 2022

PONE-D-22-05243R1 

COVID-19 mortality in the United States: It's been two Americas from the start 

Dear Dr. Stoto:

I'm pleased to inform you that your manuscript has been deemed suitable for publication in PLOS ONE. Congratulations! Your manuscript is now with our production department. 

Kind regards, 

on behalf of

Dr. Wenping Gong 

Academic Editor

PLOS ONE